# The Urban Blight Costs in Taiwan

**Chich-Ping Hu [1]**, **Tai-Shan Hu [2],\***, **Peilei Fan [3]** and **Hai-Ping Lin [2]**

1. Department of Urban Planning and Disaster Management, Ming Chuan University, Taoyuan City 333, Taiwan; chphu@mail.mcu.edu.tw
2. Department of Urban Planning, National cheng Kung University, Tainan City 701, Taiwan; s924418@gmail.com
3. Urban & Regional Planning and Center for Global Change & Earth Observations, Michigan State University, East Lansing, MI 48824-3407, USA; fanpeile@msu.edu
* Correspondence: taishan@ncku.edu.tw

**Abstract:** Urban blight is not only an eyesore for city residents, but also a threat to health, psychological well-being, and safety. It not only represents substantial economic decline, but also spreads through urban space. As well as the loss of personal property value, urban blight also harms public interests in the public domain. This study finds that danger and age are the two main factors of urban blight. Ignoring these two factors causes housing prices to fall. The decline in population due to long-term economic stagnation and the exodus of residents and industries, coupled with the long-term decline in income and spending on maintenance of old houses, has led to major visual and physical economic blight. This investigation adopts the hedonic model to analyze the correspondence of house prices with urban blight, based on real estate prices and related township variables announced by the government in Taiwan in 2017, and applies the spatial regression model to investigate the direct and indirect effects of real estate prices. The following conclusions can be drawn from the analytical results. 1. The spatial lag model finds that urban blight has a spatial spillover effect. 2. The government must not disregard the blight, due to its detrimental effect on housing prices and spatial diffusion effect. 3. The factors that affect the blight are age of residents, age of buildings, poverty, and danger.

**Keywords:** blight; hedonic model; spatial regression model

## 1. Introduction

Urban blight refers to the external costs resulting from excessive production and consumption during the long-term growth of the city, which eventually leads to market failures resulting in insufficient public facilities, old buildings, and the collapse of social security systems [1]. Unemployment and resulting long-term decline in income has caused residents to pay disproportionate maintenance costs in order to maintain the good quality of old homes. Some residents have even abandoned their properties because they cannot pay for repairs, resulting in chaos space and a substantial economic recession. Understanding the recycling of policies—both those imported from overseas and 'locally' devised responses to local problems—have been the subject of a good deal of academic attention [2]. Urban blight, with its sprawl effects, is considered a public menace to physical and mental health, and life safety of every citizen. Although the social movement that started with urban renewal in the mid-20th century has continued to reduce city deterioration for decades, neither policy makers nor social scientists have a consistent and clear definition of urban blight [3–6]. The definition and cost and benefit of urban blight varies according to the interests of different public and private stakeholders, including landlords, local government officials, builders, and citizens [5,7]. The anti-urban-blight policy in the U.S. is overseen by the Federal Economic Redevelopment Plan, and its implementation is regulated by local governments to delineate the scope of blighted areas in accordance with official needs to implement the redevelopment plan [6,8,9]. Miekley (2008) proposed five

neighborhood quantitative indicators to explain the urban blight phenomenon, namely abandoned buildings, unkempt properties, vacant lots, graffiti and litter, local crime rates, and falling property values [10]. Home buyers are willing to pay extra for a comfortable living environment, convenient location and safe living conditions.

However, city attributes representing low comfort, including old houses and elderly population (As defined by the Ministry of Interior Construction and Construction, houses over 30 years old are regarded as old houses; According to the definition of World Health Organization W.H.O., the population over 65 is regarded as the elderly population), cause real estate market transactions to shrink due to low security and low consumption levels. Urban blight factors, such as old housing, aging population, low personal income, poverty, and dangerous housing, reduce the level of housing services, and lead to a decline in demand for housing, thus reducing property prices. Comfortable city conditions are characterized by high consumer demand. Conversely, consumers lack interest in blighted housing, and are thus unwilling to buy, even at low prices [11–16]. Blight occurs when businesses and individuals leave an area of the city, thus taking jobs and property tax revenue with them [17]. Although city residents often feel that public safety is gradually worsening, poor households are forced to ignore the quality of living due to their financial instability, and the elderly are restricted and often isolated by their community. However, residents still do not feel the cost of urban blight. The blight is a hidden factor that negatively affects the operation of a city, and even the capital market. Blight is thus a hidden cost of the city, but not specifically paid by residents. The hedonic model is often used to estimate the value hidden behind the goods, especially the hidden property values of non-market goods. Non-market goods have neither market transactions nor market value, but their attributes affect the efficiency of not only the local market, but also of markets nearby. The hedonic model is often adopted to estimate the marginal value of the hidden attributes [18–20]. Some of these hidden attributes are difficult to estimate and measure, and even their implicit values are very sensitive to incomplete information [21,22].

Some attributes, such as comfort and blight, show poor information asymmetry between the buyer and the seller. In particular, this hidden attribute represents a merit or risk, increasing the complexity of estimating the marginal value of housing. To achieve a full information housing market, the model must include descriptions and estimations of the external effects of housing services [23]. The objective of this study is to investigate the cost of urban blight based on a hedonic model with property value as the dependent variable together with other independent variables of the towns and cities in Taiwan. Additionally, the spatial regression model is applied to analyze the spillover effect of blight attributes and derive the cost of urban blight by including the heterogeneity of the error term and the housing price lagged in the neighboring towns and cities as independent variables. The spatial regression model is based on spatial econometrics, which can further analyze the spatial effects from dependent as well as independent variables in terms of place. To avoid bias from estimating the marginal spatial effect, this investigation adopts the spatial hedonic model to estimate the effect of housing prices nearby neighboring towns and cities on housing prices, which cannot be analyzed by the ordinary least square (OLS) model. Government policy in Taiwan is concerned with Urban renewal, for which urban blight is the most important issue. Many studies have discussed urban renewal, and most of these have analyzed the important role of the renewal method and the transfer of rights. However, few studies have discussed the effect of urban blight on urban renewal.

## 2. Literature Review

Breger (1967) first raised urban blight issues and analyzed their causes. The article characterizes urban blight a decline in function and depreciation of value of real estate to levels unacceptable to residents of the community [7]. Morande et al. (2010) identified factors affecting urban blight as the distance between the place of residence and the nearest subway station, recoverable land after the disaster, whether the land is located in a conservation area, the population density of the community, the quality of education

in the place of residence, and neighborhood safety [24]. Brueckner and Helsley (2011) found that most residents choose to live in either downtown or suburban areas [1]. The quality and quantity of housing service in the city center depends on the level of property maintenance and reinvestment costs. However, households deciding to live in the city center must afford the external costs from a high-density living environment, unsuitable quality of life, air pollution hazards, and noise pollution disturbances caused by poor and blighted communities. The literature emphasizes that municipalities should provide sufficient information about the status of idle land and its potential in terms of ecological and social value. Create community coordinators, civic leaders, and other community-based non-profit organizations [25], will be helpful to communicate not blight. The Kuyucu and Unsal (2010) shows that the property/tenure structure of an area plays the most important role in the urban transformation projects [26]. As degree of enforcement can prompt people to engage with support services and achieve outcomes [27–38], that they themselves consider beneficial.

The article finds that overinvestment in the city center has reduced the cost of real estate acquisition in the suburban real estate market, where properties are traded lightly. The market failure in the city center pushes investment into the suburban real estate market, and suppresses the rising trend of housing prices in the city center. On the other hand, static or falling house prices in the city center reduce the incentives for homeowners to maintain their homes, and thus reduce the size of reinvestment. The article concludes that urban blight reduction can be achieved by adjusting the urban population movement policy, encouraging suburban residents to migrate into the central city, and stimulating reinvestment of the real estate market in the city center. Baum-Snow (2007) concluded that interstate highways increase the convenience of transportation among the states, but reduce the transport accessibility of city centers, and weaken the importance of the city center, leading to population decline and worsening urban blight [39]. Community crime and insufficiency of security also make urban blight worse, particularly for high-education households, and households with children, which are extremely sensitive to safety and security standards for deciding where to live. They will seriously consider moving out of communities that do not meet A [40]. Any community may experience blight.

Presently, in developing countries of Africa and Asia, urban sprawl remains a critical hurdle for urban planning and economic development of cities. Deficiencies in urban planning coupled with rapid urbanization—concentration of population in towns and cities—in developing countries have created conditions that call for immediate efforts by governments, and local agencies to respond to urban sprawl and promote healthy urbanism [41].Characteristics of inadequate housing, including long-term disrepair, over-crowding, abandoned garbage, dumping danger, loss of tax base, and higher tax burden than real estate appreciation in the community, reduce demand for housing, causing the price of housing to decline gradually, finally leading to low-income households replacing high-income households as the dominant group in the community [42]. Pritchett (2003) described the social implications of blighted areas with public threats, which cause loss of public interest in a city [43]. Valasik et al. (2018) characterized urban blight in an area as high levels of poverty, vulnerable residents, and unstable housing [33]. Blight attracts crime, further negatively affecting house prices [44–48]. A blighted community becomes a natural place for potential offenders to engage in criminal activities, because the police and the residents of the community do not care about anything that makes them feel insecure [45,49–51].

This recommends moving away from strict greenbelt containment approaches and towards flexible growth boundaries, renewing the focus on self-contained communities with a good job-housing mix, empowering local authorities to generate revenue for providing infrastructure (such as transportation and affordable housing), and achieving horizontal and vertical consistency for land-use and transport integrated planning. Measures for state transportation and land use may engender disagreement among stakeholders [52]. We argue that urban social sustainability and the overall social sustainability is a multidi-

mensional concept that incorporates six main dimensions of social interaction, sense of place, social participation, safety, social equity, and neighborhood satisfaction. Failure to consider each of these dimensions may lead to an incomplete picture of social sustainability [53]. Empirical studies have found that regeneration of urban blighted areas not only reduces the crime rate, but can also improve public health, sustain economic development, and increase real estate value [45,47,48,54,55]. In short, street vending thrives in contested public spaces amidst adverse policy environment or changing political conditions [56]. In some cities, the entrepreneurial activities associated with street vending even increase land value and generate new spatial relations [57], affecting land use and transport activities. Diversified activities in the city will promote the vigorous development of public places and reduce the urban Blight Costs.

To improve efficiency of urban policy implementation, this study recommends that researchers analyze the spatial distribution of urban blighted areas and the characteristics of the spatial distribution to refine the estimates of the cost of urban blight and its spread effects, which add to urban development cost [58,59]. Urban blight is a non-market good that lowers the individual utility and social welfare. The cost of urban blight can be estimated by indirect valuation methods, such as travel cost method and hedonic price method. The former is mostly an individual behavior model, as it requires data on travel costs and related attributes, while the latter can be applied to both individual and macro analysis [60–62]. This investigation employs a hedonic model based on aggregated data to estimate the cost of urban blight, with a spatial regression model to determine the spillover effects of blight characteristics.

## 3. Research Methods

House prices do not follow a homogeneous spatial distribution when by the ordinary least square's estimation. The factors that cause spatial variation of the distribution of housing prices are described below.

Similar spatial variables result in variation of dependent variables. Therefore, adding the independent variables to the model eliminates spatial autocorrelation in the error term by explaining the ignored independent variables.

Spatial heterogeneity causes variation. In this case, the error term still has spatial correlation, even when all the potential independent variables are added to the model. Evaluation using the least squares model is not effective. Adding regional dummy variables or independent variables may improve the effectiveness of the model.

The spatial contiguity effect means that variables of neighborhood areas influence the levels of other variables. This can be corrected by adding the dependent variable to the model as an independent variable, or by adopting the spatial lag model.

If the error term does not meet the assumption of I.I.D., then the spatial regression model can replace the least squares regression model (Figure 1). In other words, the coefficient of correlated variables is estimated with the error term assumed to be not independent and not homogeneous. Figure 1 shows the frame of the spatial regression model, which is described in detail as follows.

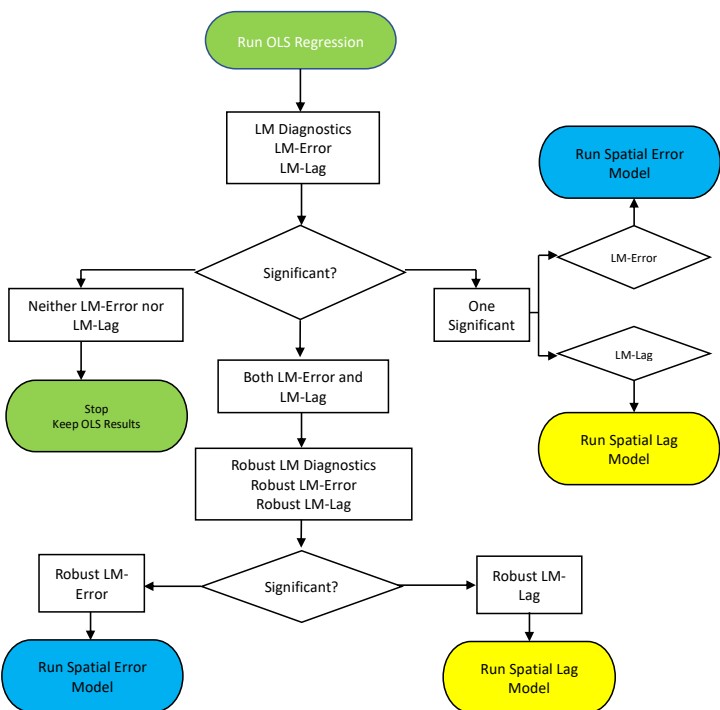

**Figure 1.** Ordinary least squares regression model and spatial regression model. Sourcel: Anselin, 2005.

The characteristic function of housing prices adopted in the least squares regression model is as Equation (1):

$$p = X\beta + u \tag{1}$$

where $p$ is the vector of housing prices in $n \times 1$ townships, cities, or districts; $X$ denotes the matrix of independent variables or the variable matrix influencing housing prices in correspondence to $n \times k$ townships, cities or districts; $\beta$ is the $k \times 1$ coefficient vector; $u$ is $n \times 1$ error term vector; $n$ represents the number of townships, cities, or districts, and $k$ stands for independent housing price variables. The error term meets the following conditions. (1) The expected value is zero and no bias exists. (2) The variance is constant and not heteroscedastic. (3) The term has no error correlation, including spatial autocorrelation. (4) The term is not endogenous, i.e., each independent variable is uncorrelated with the error term. (5) The error term, $u \sim N(0, \sigma^2)$, has a normal distribution with a mean of zero and variance of $\sigma^2$.

Moran's $I$, proposed by Cliff and Ord (1972), is the most popular measurement for the spatial autocorrelation with the null hypothesis $H_0$ and the error term of IID when analyzing global spatial autocorrelation. Equation (2) displays Moran's $I$ calculation for global housing prices.

$$I = \frac{n}{\sum_{i=1}^{n}\sum_{j=1}^{n}W_{ij}} \times \frac{\sum_{i=1}^{n}\sum_{j=1}^{n}W_{ij}(p_i - \overline{p})(p_j - \overline{p})}{\sum_{i=1}^{n}(p_i - \overline{p})^2} \tag{2}$$

where $I$ denotes the statistic of global spatial autocorrelation; $p_i$ denotes the mean of housing prices in township, city, or district $i$, and $\overline{p}$ denotes the mean of housing prices in all townships, cities, and districts in Taiwan. $W_{ij} = 1$ if township, city, or district $i$ is adjacent to township, city, or district $j$, and $W_{ij} = 0$ otherwise (the township, city, or district i and the township, city, or district j adjacent to each other indicate that they have a common boundary, otherwise the two are not adjacent). Global spatial autocorrelation is the index of housing prices, showing whether the spatial distribution is clustered, dispersed, or random, although it cannot specifically indicate the cluster location.

The index of local spatial autocorrelation further demonstrates the characteristics of housing prices in spatial distribution, when performing Getis-Ord for the analysis of local

spatial autocorrelation, $G_i^*$. The local spatial autocorrelation in township, city, or district $i$ is calculated by Equation (3):

$$G_i^*(d) = \frac{\sum_{j=1}^n W_{ij}(d)p_j}{\sum_{j=1}^n p_j}, j = i \tag{3}$$

$p_j$ denotes the mean housing price in township, city, or district $j$; $i$ and $j$ are adjacent towns or cities, and $W_{ij} = 1$ if the distance from $i$ to $j$ is less than $d$. Two towns or cities with distance greater than $d$ apart are not considered adjacent. That is, $W_{ij} = 0$.

Normalization of $G_i^*$ and significance testing are conducted. Equation (4) illustrates the calculation of $Z(G_i^*)$:

$$Z(G_i^*) = \frac{G_i^* - E(G^*)}{\sqrt{V(G^*)}} \tag{4}$$

$E(G^*) = \frac{\sum_{i=1}^n \sum_{j=1}^n W_{ij}}{n(n-1)}, \forall j \neq i$ denotes the mean of the index of local spatial autocorrelation, $G_i^*$, for housing prices in all townships, cities, and districts. $V(G^*) = E\left(G^{*2}\right) - E(G^*)^2$ stands for the variance for $G_i^*$. If $Z > 1.65$, then $p < 0.10$, indicating that the hot zone of housing prices is within the 90% confidence level. For $Z > 2.81$, $p < 0.01$, which means that the hot zone of housing prices is within the 99% confidence level. For $Z < -1.65$, $p < 0.10$. In this case, the cold zone of housing prices exists within the 90% confidence level. For $Z < -1.96$ and $Z < -2.81$, $p < 0.05$ and $p < 0.01$, meaning that the cold zone of housing prices lies within the 95% and 99% confidence level, respectively.

Based on Anselin (1988), the spatial regression model is adopted to build a hedonic price model for investigating the cost of urban blight [63]. The Lagrange Multiplier is adopted to test the spatial lag in housing prices $Wp$ and Hypothesis $H_0$ (adding the housing price as an independent variable does not increase the appropriateness of model). The Lagrange Multiplier is also adopted to test the spatial lag for the error term $W_u$ and $H_0$ (adding the error term as independent variable does not make the appropriateness of model better). If neither of these two spatial lag values are significant, then the least-squares method is adopted. If the spatial lag for $W_u$ and $H_0$ is significant, then the spatial error model is selected and described as Equation (5).

$$\begin{aligned} p &= X\beta + u \\ u &= \lambda Wu + \epsilon \end{aligned} \tag{5}$$

$\lambda$ denotes the spatial regression coefficient; $W$ denotes a spatial weighted matrix of order $n \times n$, and $\epsilon$ is the vector of error term. If the spatial lag of $Wp$ and $H_0$ is significant, then Equation (6) is used.

$$p = \rho Wp + X\beta + u \tag{6}$$

$Wp$ denotes the spatial lag term, and $\rho$ denotes the spatial regression coefficient. If both these variables are significant, then robustness testing is performed.

(i) Roughness testing for spatial error model. $H_0$: $Wu$ and $Wp$ both exist in the model. The appropriateness of the model remains the same after removing $Wu$. (ii) Roughness testing for spatial lag model. $H_0$ : $Wu$ and $Wp$ both exist in the model. Removing $Wp$ does not change the appropriateness of the model. If (i) is significant, then the spatial error model is chosen. If (ii) is significant, then the spatial lag model is applied.

## 4. Results and Discussion

### 4.1. Research Variables

The data displayed by categorization of 368 townships in Taiwan, cities, and districts shows overall information. The overall framework that affects the development of urban housing is discussed (internal/external), and the research mainly focuses on the analysis of external "Urban Blight Costs" conditions that affect housing. First, the areas of urban blight were identified from the definitions and variables described in the references, as well

as considering the data availability and suitability announced by township, city, or district government. Areas of urban blight, by definition, include those with old houses, elderly population, declining income, increased risk, and dead zones. Blight spreads, and is a hidden cost to society. The spread of social cost becomes a spatial spillover effect. Define *A* and *B* as adjacent towns or cities, where τ denotes the marginal effect of house age to the housing price in area *A*, and ρ denotes the spillover effect of housing prices from area *B* to area *A*. The effect of house age on house price is derived from the recursive function. First, if τ denotes the marginal effect of house age to the housing price in area *A*, then the housing price in area *A* affected by τ results in ρ, influencing the housing price in the adjacent area *B*. The fall in housing prices in area *B* further leads to the spillover effect τ of housing prices in the area *A*, and again spills over to area *B* until the spillover effect reaches convergence.

Information about different townships, cities, and districts in Taiwan in 2017 was obtained through the system of Ministry of Finance and Ministry of the Interior. To establish a digital map, the data were also input into a geographic transformation system, ArcGis. Taiwan, located on the border of the Ring of Fire, the Eurasian Plate, and the Philippine Sea Plate, encounters earthquakes frequently. Thus, houses built with brick, wood, or stone are considered fragile buildings. Buildings aged more than 50 years are risky and fragile regardless of earthquakes, flooding, or public safety. The variables adopted in this investigation. Although household income is a global variable, it affects the housing services and the decision to rent or buy a house from the perspective of individual behavior. The factors of urban blight affect housing prices. The quarterly housing price, regardless of housing type, reflects the social cost caused by urban blight. A blighted area reveals the failure of a place, and implies an aging population not associated with spatial elements (Table 1). The mean household income is NT$791,000. The number of buildings built with brick, wood, or stone is 1889. The numbers of houses aged over 50 years and under 1 year are 1726 and 270, respectively. The elderly population per household is 0.45 people. The mean low-income population is 862.14 persons. The mean housing prices are NT$8,867,000, NT$11,618,000, NT$8,848,000, and NT$9,041,000 from Quarter 1 to Quarter 4, respectively.

**Table 1.** Variables adopted and descriptive statistics of the analysis of urban blight effects in townships, cities, and districts in Taiwan in 2017.

| Variable | Definition of Blight | Description | Unit | Source | Mean | Standard Deviation | Minimum | Maximum |
|---|---|---|---|---|---|---|---|---|
| Income | | Household income | Thousand Dollars | Financial Data Center, Ministry of Finance | 791.3243 | 166.3347 | 591.9725 | 1734.0120 |
| Special Stone House | Danger | Building material of brick, wood, or stone | House | Real Estate Information Platform, Ministry of the Interior; REIP, MOI | 1889.016 | 1390.542 | 41 | 15,018 |
| Old House | Oldness | House aged over 50 years | House | REIP, MOI | 1725.626 | 1445.710 | 3 | 10,247 |
| Young House | | House aged under 1 year | House | REIP, MOI | 270.4478 | 534.3672 | 0 | 4645 |
| House Age | | Average house age | Year | REIP, MOI | 33.8481 | 6.4780 | 15.71 | 66.55 |
| Senior | Seniors | Elderly population per household | Person | REIP, MOI | 0.4461 | 0.1141 | 0.20 | 0.78 |
| Low Income | Poverty | Low-income population | Person | Ministry of Health and Welfare | 862.1423 | 1095.527 | 2 | 7629 |
| Housing Price | | Average housing price listed on the purchase agreement (regardless of housing type) | Ten Thousand Dollars | REIP, MOI | 904.0564 | 434.9548 | 273.7 | 3035 |

Source: "Financial Data Center, Ministry of Finance" and "Real Estate Information Platform, Ministry of the Interior".

*4.2. The Analysis of the Hot Zone, Cold Zone, and Variables Correlated with Geographic Distribution*

A geographic information system visually depicts the geographic distribution of variables. The spatial econometric model can analyze econometric indices, and visualize spatial structures. Researchers can identify the characteristics of spatial distribution, such as housing prices and the related variables, based on the map generated by the model. The spatial clustering model can be further applied to identify the hot zone, cold zone outlier and the ones not spatially related, based on the distribution of variables (Table 2, Figure 2). This investigation discusses spatial distribution, the hot zone, and the cold zone based on eight indices. The analysis indicates that the hot zone of high housing prices in New Taipei City is characterized by distribution of housing prices, household income, Special Stone House, Old House, or Young House, Senior. The hot zone in Taoyuan City is characterized by distribution of high housing prices, distribution of housing prices, Income, and Young House. Conversely, distribution of housing prices, Income, and Young House form the cold zone in the central to southern (Chiayi City and County, Yunlin County, and Tainan City) and east coast of Taiwan (Hualien County and Taitung County). The hot zone in these regions, (especially OR particularly) in Chiayi County, Tainan City, and Taitung County, is characterized by Special Stone House, Old House, average house age, and Seniors.

Hence, this investigation defined four factors as Seniors, house age, poverty, and danger, by categorizing the indexes of urban blight, and further conducted cross-analysis of the factors. This article uses the data of the "low-income households" of the Ministry of Health Service, the "number of houses with a house age of more than 50 years", "the number of brick, wood, and stone houses", and the "average number of elderly people per household" of the real estate information platform of the Ministry of the Interior. By standardization of the four variables, they represent "poverty", "oldness", "danger", and "seniors". The sum of standardized z values can have "oldness" and poverty", "oldness and seniors", "oldness, poverty, and danger", "oldness, seniors, and poverty", "oldness seniors, poverty, and danger" variables in different combinations. The intersection of elderly population and poverty shown in Figure 2a form a hot zone in central Taiwan (Yunlin County and Chiayi County) and a cold zone in southern Taiwan (Pingtung County). Figure 2c illustrates the hot zone of New Taipei City, Yunlin County, and Nantou County, and the cold zone of Taoyuan City and Tainan City, concerning the intersection of Seniors and House Age. The spatial distribution for the intersection of Seniors, House Age, and Danger forms a hot zone in central to southern Taiwan, including Yunlin County, Tainan City, Nantou County, and Chiayi County (Figure 2d), and a cold zone in New Taipei City in northern Taiwan. The factors of Seniors, House Age, and Poverty form hot zones in Yunlin County, Chiayi County, and Tainan City, and a cold zone in Pingtung County (Figure 2e). Figure 2f illustrates the result of analysis combining the factors of Seniors, House Age, Poverty, and Danger. Spatial distribution results indicate that these factors form a hot zone in Yunlin County, Chiayi County, Tainan City, and Taitung County, and a cold zone in Hualien County.

Spatial autocorrelation investigation results demonstrate that the blighted characteristics not only exist in urban areas, but are also distributed in rural areas, and have spatial concentration.

**Table 2.** The geographic distributions, hot zones and cold zones of housing price, blights, and relative variables in townships, cities, and districts in Taiwan.

| Variable | (Hot Zone) | (Cold Zone) |
|---|---|---|
| High housing price | New Taipei City, Taoyuan City, Hsinchu City, Taichung City | - |
| Housing prices (Only the 99% is considered for the cold zone.) | Yilan County, New Taipei City, Taipei City, Keelung City, Taoyuan City, Kinmen County | Taitung County, Hualien County, Pingtung County, Kaohsiung City, Tainan City, Chiayi County, Yunlin County |
| Household income (Only the 99% is considered for the cold zone.) | Yilan County, Kinmen County, Taoyuan City, Keelung City, New Taipei City, Hsinchu County, Taipei City | Hualien County, Nantou County, Pingtung County, Kaohsiung City, Yunlin County, Chiayi County, Taitung County, Tainan City |
| Danger | Yilan County, Hualien County, Taoyuan City, Kaohsiung City, New Taipei City, Taitung County, Tainan City, Penghu County | Kaohsiung City, Yilan County, Taoyuan City, New Taipei City, Hualien County |
| Oldness | Penghu County, New Taipei City, Taoyuan City, Yilan County | Taitung County, Kaohsiung City |
| Young House | New Taipei City, Taoyuan City | Hualien County, Kaohsiung City, Taitung County, Tainan City |
| House age | New Taipei City, Taoyuan City, Yunlin County, Changhua County | Kaohsiung City |
| Senior | Hualien County, Nantou County, Taoyuan City, Kaohsiung City, Yunlin County, New Taipei City, Chiayi County, Changhua County, Taichung City, Taitung County, Tainan City, Penghu County | Kaohsiung City, Taoyuan City, Taitung County |
| Seniors and poverty | Nantou County, Changhua County, Taichung City, Penghu County | Taitung County, Penghu County |
| Seniors and oldness | Penghu County, Taichung City, Keelung City, Changhua County | Taitung County, Kaohsiung City, Taichung City, Taoyuan City, New Taipei City |
| Seniors, oldness and danger | Hualien County, Nantou County, Kaohsiung City, Yunlin County, Changhua County, Taitung County, Tainan City, Penghu County | Taitung County, New Taipei City, Taoyuan City, Yilan County |
| Seniors, oldness and poverty | Nantou County, Yunlin County, Changhua County, Taichung City, Penghu County | Taitung County, Pingtung County |
| Seniors, oldness, poverty and danger | Yunlin County, Changhua County, Tainan City, Penghu County | Taitung County |

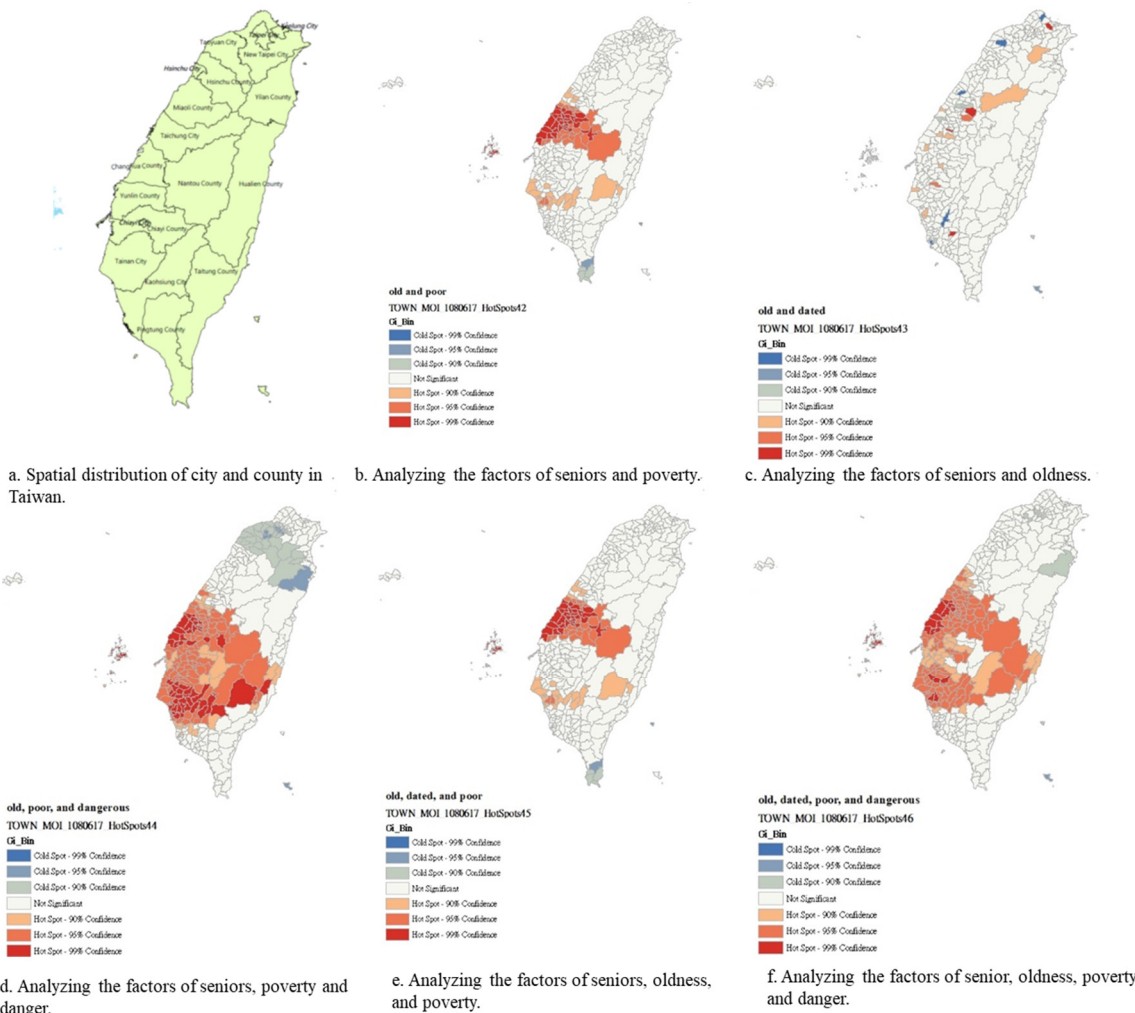

**Figure 2.** The hot zone of spatial distribution of Taiwan in 2017 by Getis-Ord.

### 4.3. Factors Causing Urban Blight and the Economic Costs

Seniors, House Age, poverty, and danger are the key factors causing urban blight, which reduce housing prices, further causing spatial spread. This investigation established a characteristic function for housing prices, selecting the median housing prices from Quarter 1 to Quarter 4 of 2017 in Taiwan as the dependent variables. The dependent variables comprised the Income, Special Stone House, Old House, Young House, Senior, and Low Income. These variables are the major factors that cause urban blight.

First, the characteristic function of housing prices was established by the least squares model (Table 3). Income and Low Income were the two factors that most significantly influenced housing prices. The estimated coefficients were positive, indicating that the result was incompatible with factor (the Variance Inflation Factor (VIF) of the independent variable is lower than 10, indicating that there is no collinearity between the variables) of urban blight. The overall model did not pass the characteristic test for independent and identically distributed error terms (independent and identically distribution test for the error term of housing price of 2017 in Taiwan. H0: Error term is I.I.D.; Errorlags: W; $\chi^2$ (1) = 35.52; Prob.$\geq \chi^2$ = 0.0000), possibly because of the characteristics of spatial clustering for housing prices or the spillover effect on the housing prices in a certain space range.

**Table 3.** Estimation results of housing prices function in Taiwan by ordinary least squares method for 2017.

| Housing Price [a] | Coefficient | Standard Deviation | VIF |
|---|---|---|---|
| Constant Term | −479.6246 | 250.7130 | |
| Income | 1.3936 *** | 0.1378 | 1.59 |
| Special Stone House | −0.0437 | 0.0232 | 1.95 |
| Old House | −0.0078 | 0.0198 | 3.04 |
| Young House | −0.0258 | 0.0431 | 1.79 |
| House Age | −0.8420 | 10.9863 | 5.38 |
| Senior | 312.892 | 597.5865 | 4.29 |
| Low Income | 0.1046 *** | 0.0200 | 1.72 |
| Adj $R^2$ | 0.6149 | | |
| Prob. >F | 0.0000 | | |
| Mean VIF | 2.82 | | |

Note: ***: $p < 0.001$; [a]: Median of housing prices from Quarter 1 to Quarter 4 of 2017.

To avoid statistical errors, this investigation also conducted normality, heterogeneity, and spatial dependence tests for the error term[4]. The displays the results of Jarque–Bera Test for normality test within 1% significance level ($p = 0.00$, rejecting $H_0$). The error term was not normally distributed. Regarding the heterogeneity test, the test result of the Breusch–Pagan Test is 174.652, whereas that for Koenker–Bassett Test showing that the variance was not stationary, nor a fixed constant. Possible root causes for the heterogeneity of housing prices in Taipei City are accessibility of the location, completeness of infrastructure, and external economy. Therefore, this investigation adopted the Lagrange Multiplier for test of dependence and added variables. Significant influence by a certain variable indicated the dependence and importance of that variable.

Analytical results indicate that both the spatial error model and spatial lag model were more suitable than the least squares model. Therefore, robustness tests for both spatial error and spatial lag models were undertaken. The robust spatial lag model 4 demonstrated better suitability than the Lagrange Multiplier model. Robust LM (Error) random variable test value is 3.216 with a p value of 0.0729, accept hypothesis $H_0$: remove error term independent variable, the model's fitness is not reduced to indicate that the spatial error model estimation is inferior to the ordinary least-squares estimation. However, its fitness is lower than that of the Lagrange Multiplier estimation method. Robust LM (Lag) random variable test value is 8.503 with a p value of 0.0035, reject hypothesis $H_0$: remove house prices as independent variables, the model is properly reduced to indicate that the spatial lag model estimation is better than the ordinary least squares estimation. This investigation adopted the robust spatial lag model to analyze the marginal utility for urban blight.

A. Spatial error model and spatial lag model for housing prices

The error term is not independent and identically distributed when estimating housing prices by least squares estimation. The shows results from a further Lagrange Multiplier test, which indicates that both models passed the test. Accordingly, this investigation conducted a robustness test. The significance level increased for the spatial error model, as demonstrated by the increase in p value from 0.0086 to 0.0729. The significance level for spatial lag model diminished, as demonstrated by the decrease in p value from 0.0081 to 0.0035. That is, the spatial lag model has a lower significance level of than the spatial error model.

B. The robust spatial error model and robust spatial lag model for housing prices

Both spatial regression models passed the Lagrange Multiplier test. Therefore, robustness test 5 was further conducted. The p values for the spatial lag model and spatial error model in Independent and identically distribution test for the error term of housing price of 2017 in Taiwan are 0.0035 and 0.0729, respectively. However, only the spatial lag model rejected $H_0$. Robust spatial lag model was adopted for the analysis of spatial effect

on housing prices. This investigation incorporated the robust spatial lag model for the analysis of spatial effect on housing prices. The significant variables listed in Table 4 are the Income, Special Stone House, Young House, and Low Income.

C.    The spatial characteristics of urban blight influencing housing prices

The characteristic function for blight influencing housing prices was established based on the robust spatial regression model. The estimated regression coefficients were further applied to calculate the marginal utility for each blight characteristic. A negative value of marginal utility for a characteristic in Table 4 indicates that the characteristic decreased the housing price, and the price of a blight characteristic represents its contribution to the overall cost linked to blight.

i.    A decrease of average household income (Income) by NT$1000 directly caused housing prices to drop by NT$13,200. The spillover effect of the adjacent areas decreased the housing price by NT$900. The total effects on housing prices was thus a decrease of NT$14,100.

ii.    One additional house built with brick, wood, or stone (Special Stone House) directly lowered the housing prices by NT$400. Housing prices further dropped by NT$30 owing to the indirect effect resulting from the spillover effect of the neighboring areas. The total effect was to reduce the housing prices by NT$430.

iii.    Increasing one house aged more than 50 years (Old House) led to a drop in housing prices of NT$140. The indirect effect caused by the spillover of the adjacent areas was to reduce housing prices by NT$10. The total spatial effect was a reduction in housing prices of NT$150.

iv.    Housing prices declined by NT$680 when adding one house aged less than one year (Young House). The spillover of adjacent areas indirectly reduced the housing prices NT$50. The overall effect was thus a decline in housing prices of NT$730.

v.    An additional one year of house age (House Age) directly reduced the housing prices by NT$65,790. The spillover effect of the neighboring areas indirectly lowered the housing prices by NT$4670, resulting in a drop of NT$70,460 in the overall housing prices.

vi.    Reducing the average senior population per household by 1 person (Senior) directly lowered the housing prices by NT$4,477,990. The indirect effect of the spillover of adjacent areas was a NT$318.06 drop in housing prices. The overall effect was thus a decrease in housing prices of NT$4,796,050.

vii.    The low-income population (Low Income) decreasing by one person directly lowered housing prices by NT$1,040. The spillover effect of the adjacent areas caused a reduction of housing prices by NT$70. Thus, total effect was a reduction in the housing prices NT$1,120.

Based on the above description, the blight characteristics that meet the requirement of negative values of marginal utility are Special Stone House, Old House, Young House, and House Age. In contrast, Income, Senior, and Low Income gave a positive value of marginal utility.

**Table 4.** Estimation results of spatial regression model and Spatial effect of housing prices of 2017 in Taiwan.

| Housing Price [a] | Spatial Error Model | | Spatial Lag Model (Robust) | | Spatial Effect with Respect to Changes by the Relative | | |
|---|---|---|---|---|---|---|---|
| | Coefficient | Standard Deviation | Coefficient | Standard Deviation | Direct Effect | Indirect Effect | Total Effect |
| Constant Term | −187.7682 | 261.0479 | −358.3282 | 244.6073 | 1.3242 | 0.0941 | 1.4182 |
| Income | 1.3179 *** | 0.1405 | 1.3219 *** | 0.1345 | −0.0400 | −0.0028 | −0.0428 |
| Special Stone House | −0.0331 | 0.0235 | −0.0399 | 0.0220 | −0.0141 | −0.0010 | −0.0151 |
| Old House | −0.0112 | 0.0216 | −0.0141 | 0.0220 | −0.0683 | −0.0049 | −0.0732 |
| Young House | −0.0908 * | 0.0330 | −0.0682 | 0.0403 | −6.5789 | −0.4673 | −7.0462 |
| House Age | −8.9127 | 10.8185 | −6.5675 | 10.8401 | 447.799 | 31.806 | 479.605 |
| Senior | 350.2758 | 562.6866 | 447.0207 | 570.4571 | 0.1041 | 0.0074 | 0.1115 |
| Low Income | 0.1063 *** | 0.0192 | 0.1039 *** | 0.0194 | 1.3242 | 0.0941 | 1.4182 |
| w Housing Price 4 | | | 0.1184 * | 0.0438 | | | |
| e. Housing Price 4 | 0.3410 *** | 0.0628 | | | | | |
| Var(e. Housing Price 4) | 64719.92 | 7399.522 | 65594.32 | 7368.012 | - | | |
| Pseudo $R^2$ | 0.6343 | | 0.6403 | | | | |
| Wald $\chi^2$ (Prob. $>\chi^2$) | 6.902(0.0086) | | 7.011(0.0081) | | | | |
| AIC | 2290.785 | | 2290.727 | | | | |
| BIC | 2321.665 | | 2321.661 | | | | |

Note: ***: $p < 0.001$, *: $p < 0.01$; [a]: Median of housing prices from Quarter 1 to Quarter 4 of 2017.

## 5. Conclusions and Suggestion

The cost of urban blight includes deindustrialization, population decline, deurbanization, economic reconstruction, abandoned residences and public facilities, high unemployment rate, poverty, family disintegration, low standard of living, low life quality, political deprivation, crime, increased pollution, and desolate urban landscapes. However, residents usually do not feel the cost because they do not directly pay for it. However, those hidden environmental costs lower the prices of real estate. This investigation first explains the variable coefficients through the least-squares regression model, and indicates that only two factors, Income and Low Income, were significant. That is, the model was unable to explain other variables. Although the spatial collinearity was not significant, the indicator of the global spatial autocorrelation for housing prices, Moran's I, and that for local spatial autocorrelation, Getis-Ord $G_i^*$, both indicated that the error term of the estimation did not correspond to the I.I.D. assumption. To avoid any bias, this investigation adopted the spatial economic model for the estimation of housing prices. Results of the suitability test for the models indicate that the spatial lag model is most suitable for the estimation of housing prices.

Besides the ordinary least-squares estimation, spatial autocorrelation, spatial heterogeneity, and abnormality were identified in the error term of the adopted model. Bias in the estimation results from spatial heterogeneity of the housing prices; the relatively high housing prices in Taipei area and the areas adjacent to Taipei; the relatively low housing prices in Ping-tong, Hualien, Taitung, and the adjacent areas; high and low housing prices accompanying spatial clustering, and the huge differences between high and low housing prices. Furthermore, this investigation revealed that factors of Income, Low Income, and Senior did not meet the characteristics of urban blight. Conversely, Special Stone House, Old House, Young House, and House Age contributed to urban blight. In general, the spillover effect occurred on the housing prices, influencing the housing prices not only in an environment but also in the adjacent areas. The affected housing prices in the adjacent areas would further influence the housing prices in the original area. The housing prices would be affected recursively until the spillover effect reached convergence.

The spatial hedonic model constructed in this study is a generalized model, with variables taken from official databases, so can be applied generally to policy analysis in other regions. The explained variable is the average urban housing price, and the explanatory variables are mainly housing-related attributes.

**Author Contributions:** C.-P.H. initiated the project and conducted the analysis. T.-S.H., and H.-P.L. helped the design of research framework. C.-P.H., T.-S.H., H.-P.L., and P.F. collaborated on drafting the manuscript and multiple revisions. All authors have read and agreed to the published version of the manuscript.

**Funding:** The Ministry of Science and Technology of Taiwan for partially financial supporting this research under Contract Numbers MOST 109-2221-E-130-001-.

**Acknowledgments:** The authors would like to thank the Ministry of Science and Technology of Taiwan for partially financial supporting this research under Contract Numbers MOST 109-2221-E-130-001-. The authors are also grateful to the anonymous reviewers who provided useful comments on an earlier draft of the paper.

**Conflicts of Interest:** The authors declare no conflict of interest.

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
