# Peer review of "The Urban Blight Costs in Taiwan"

_sustainability, doi:10.3390/su13010113_

Round 1

Reviewer 1 Report

Abstract

The abstract needs some improvements. The research goal and its context should be framed. A better description of the methodology is necessary, and the main result should be identified. Two of the five conclusions must be removed to achieve a better summary.

1-Introduction

The research question does not appear clearly evidenced for the reader to understand the reason for the study. The scientific term and studies of “urban blight” are not sufficiently explained according to the scientific bibliography presented.

The main objective is not clearly presented, and the new, different and innovative approach that brings the study to the scientific community in the field of sustainability is not understandable. The introduction section is not clearly structured.

Line 68-70, What is described and the need for an urban model are not sufficiently justified.

2-Literature Review

The manuscript as it is written (introduction and Literature Review), directs the reader to a particular position on urbanism, and does not allow a free reflection of knowledge. The authors appear to draw up a previous judgment of value about urban development without presenting clear data.

At the end of “Literature review”, line 159, the main reason for the research and its contribution should be presented.

3-Research Methods, 4-Empirical Research

There is no results and discussion section, only a reference to Analytical results, line 326. This invalidates the understanding, reading and discussion of the research.

Figures 1 and 2 appear described in the text but are not located in it. The same for the tables.

References

35% of the bibliographic references are fifteen or more years old, which the authors should present more recent ones.

Author Response

Response to Reviewer 1 Comments

Point Abstract: The abstract needs some improvements. The research goal and its context should be framed. A better description of the methodology is necessary, and the main result should be identified. Two of the five conclusions must be removed to achieve a better summary.

Response Abstract: Thank you for your suggestion. This paper has referenced your comments to modify the content of the abstract, and has removed two of the conclusions. Please refer to line 5-8 and line15-18. It not only represents substantial economic decline, but also spreads through urban space. As well as the loss of personal property value, but urban blight also harms public interests in the public domain. This study finds that danger and age are the two main factors of urban blight…

Point 1-Introduction: The research question does not appear clearly evidenced for the reader to understand the reason for the study. The scientific term and studies of “urban blight” are not sufficiently explained according to the scientific bibliography presented.

The main objective is not clearly presented, and the new, different and innovative approach that brings the study to the scientific community in the field of sustainability is not understandable. The introduction section is not clearly structured.

Line 68-70, What is described and the need for an urban model are not sufficiently justified.

Response 1-Introduction: Thank you for your comments. This paper has referred to your suggestions to illustrate the research question and explain the meaning of “urban blight”. Please refer to line 21-23. Urban blight refers to the external costs resulting from excessive production and…

Thank you for your suggestions. This paper has referred to your comments to illustrate the research objective. Please refer to Line 68-70. For the purpose of clear statement of the research objective, supplementary description Line 44-52 is added.However, city attributes representing low comfort, including old houses and elderly population, cause real estate market transactions to shrink due to low security and low consumption levels...

Thank you for your comments. This paper has referred to your comments to supplement the research objective. Please refer Line 68-70, and line 75-78.

Point 2-Literature Review: The manuscript as it is written (introduction and Literature Review), directs the reader to a particular position on urbanism, and does not allow a free reflection of knowledge. The authors appear to draw up a previous judgment of value about urban development without presenting clear data.

At the end of “Literature review”, line 159, the main reason for the research and its contribution should be presented.

Response 2-Literature Review: Thank you for your comments. This paper has added relevant literature based on your comments. Please refer to Line100-103, Line156, and Line160-161. In order to avoid misunderstanding the research structure, this paper has added the description of Line 161-163.

Thank you for your comments. This paper has added relevant literature based on your comments. Please refer to Line169-171.

Point 3-Research Methods, 4-Empirical Research: There is no results and discussion section, only a reference to Analytical results, line 326. This invalidates the understanding, reading and discussion of the research.

Figures 1 and 2 appear described in the text but are not located in it. The same for the tables.

Response 3-Research Methods, 4-Empirical Research: Thank you for your comments. This paper has referred to your suggestions for supplementary descriptions and explanations. Please refer to Line 325-333.

Thank you for your comments. The paper has marked the location of the pictures and tables in the text. Line 244, 311 and 384.

Point References: 35% of the bibliographic references are fifteen or more years old, which the authors should present more recent ones.

Response References: Thank you for your comments. This paper has been supplemented with 10-13 papers in the past five years by referring to your instructions. Please refer to the content in red font from Line 418 to Line 569.

Reviewer 2 Report

This paper may be published after a major revision.

The abstract of the paper is good.

"Urban blight" deserves a better definition at the begin. Also, there are questionable affirmations, such as overall stagnation of economic growth of cities. Does this happen everywhere?

The paper is generally well organised, with some shortcomings. The Results and discussion section is not properly represented.

It is good to present the mathematics behind the method. More reference to MCDM and which standard method lies closest is needed.

The Results could be better presented, including some maps with the affected areas, to support the geography which might be relevant in the research. This should be properly included in the discussion.

Finally, some generalisation possibilities should be shown. How could a similar research be applied in other locations? How should costs be adapted?

The format of the paper is also not corresponding to the journal.

Author Response

Response to Reviewer 2 Comments

Point: This paper may be published after a major revision. The abstract of the paper is good.

Response: Thank you for your affirmation of this paper.

Point: "Urban blight" deserves a better definition at the begin. Also, there are questionable affirmations, such as overall stagnation of economic growth of cities. Does this happen everywhere?

Response: Thank you for your comments. This paper has referred to your suggestions for supplementary explanation. Please refer to Line 21-23.

Point: The paper is generally well organised, with some shortcomings. The Results and discussion section is not properly represented.

Response: Thank you for your comments. This paper has referenced your suggestions to strengthen the description of results and discussion. Please refer to Line 224-225, and 309-310.

Point:It is good to present the mathematics behind the method. More reference to MCDM and which standard method lies closest is needed.

Response: Thank you for your suggestions. This paper has referred to your comments to illustrate the research objective. Please refer to Line 51-52. For the purpose of clear statement of the research objective, supplementary description Line 68-70 is added. Also For the methods of clear statement of the research objective, supplementary description Line 224-225.

Point: The Results could be better presented, including some maps with the affected areas, to support the geography which might be relevant in the research. This should be properly included in the discussion.

Response: Thank you for your comments. This paper has referred to your suggestions for supplementary descriptions and explanations. Please refer to Line 295-308.

Point: Finally, some generalisation possibilities should be shown. How could a similar research be applied in other locations? How should costs be adapted?

Response: Thank you for your suggestions. This paper has referred to your suggestions for supplementary descriptions and explanations. Please refer to Line 411-414.

The spatial hedonic model constructed in this paper is a generalized model, and the variables are taken from official databases. It can be generally applied to policy analysis in other regions...

Point: The format of the paper is also not corresponding to the journal.

Response: Thank you for your comments. This paper has referred to your suggestions for supplementary descriptions and explanations.

Reviewer 3 Report

Dear Authors, the article is clear and well structured.

It's very technical; it's fine but it could be usefull to make it "soften" with some more reflection (not much more then few lines to add in the text).

I don’t see references about the International Valuation Standard; in these standard there are some references to regression model for the appraisal of real estate assets. It may be interesting to refer to these standards to contextualize the assessment approach you used in the article.

To increase the readability (as I said the article is very technical) I suggest to explaine that the value of a real estate asset depend on endogenous (which concern the asset) and exogenous (which you correctly name “city characteristics”) components; it will be usefull in paragraph 2, to expand what you said in line 43 about city characteristic. A brief insight above connections about urban blight and specific shortcomings could give the reader further elements for reflection.

If you want I suggest you 2 articles where you can find some criteria and indicators about quality in an urban area (opposite of urban blight):

Guarini M.R., Battisti F. (2014). Benchmarking multi-criteria evaluation methodology’s application for the definition of benchmarks in a negotitation-type public-private partnership. A case of study: the integrated action programmes of the Lazio Region. In: International Journal of Business Intelligence and Data Mining, Vol. 9, No. 4, pages 271 – 317. DOI: 10.1504/IJBIDM.2014.068456

Battisti F., Guarini M.R. (2017). Public interest evaluation in negotiated public-private partnership. International Journal of Multicriteria Decision Making, 7(1), pages 54-89. DOI: 10.1504/IJMCDM.2017.10006053

Pay attention to the format of the article starting from line 331.

Best Regards

Author Response

Response to Reviewer 3 Comments

Point: Dear Authors, the article is clear and well structured..

Response: Thank you for your affirmation of this paper.

Point: It's very technical; it's fine but it could be usefull to make it "soften" with some more reflection (not much more then few lines to add in the text).

Response: Thank you for your comments. This paper has referred to your suggestions for supplementary descriptions and explanations. Please refer to Line 300-310, and 316-318.

Point: I don’t see references about the International Valuation Standard; in these standard there are some references to regression model for the appraisal of real estate assets. It may be interesting to refer to these standards to contextualize the assessment approach you used in the article.

Response: Thank you for your comments. This paper has added relevant literature based on your comments. Please refer to Line 99-103, Line156-158, and Line161-163.

Point: To increase the readability (as I said the article is very technical) I suggest to explaine that the value of a real estate asset depend on endogenous (which concern the asset) and exogenous (which you correctly name “city characteristics”) components; it will be usefull in paragraph 2, to expand what you said in line 43 about city characteristic. A brief insight above connections about urban blight and specific shortcomings could give the reader further elements for reflection.

Response: Thank you for your comments. This paper has referred to your suggestions for supplementary descriptions and explanations. Please refer to Line 254-257.

Point: If you want I suggest you 2 articles where you can find some criteria and indicators about quality in an urban area (opposite of urban blight):

Guarini M.R., Battisti F. (2014). Benchmarking multi-criteria evaluation methodology’s application for the definition of benchmarks in a negotitation-type public-private partnership. A case of study: the integrated action programmes of the Lazio Region. In: International Journal of Business Intelligence and Data Mining, Vol. 9, No. 4, pages 271 – 317. DOI: 10.1504/IJBIDM.2014.068456

Battisti F., Guarini M.R. (2017). Public interest evaluation in negotiated public-private partnership. International Journal of Multicriteria Decision Making, 7(1), pages 54-89. DOI: 10.1504/IJMCDM.2017.10006053

Response: Thank you for your comments. This paper has referred to your suggestions for supplementary descriptions and explanations. Please refer to Line 161-163.

Point: Pay attention to the format of the article starting from line 331..

Response: The format is incorrect, review it again and make corrections in accordance with the journal format.

Round 2

Reviewer 1 Report

Dear Authors

Thank you for the improved version of the manuscriptr, as well as its changes, adjustments and clarifications on some points throughout the text. This version is clearer for the reader.

Sincerely

Author Response

Response to reviewer 1

Thank you for your suggestion and affirmation.
We will confirm the format of the article to the journal editor.

All the best.

Reviewer 2 Report

The paper improved adequately, however, the figures and tables shall be inserted in the manuscript for proper viewing.

Author Response

Response to Reviewer 2

Thank you for your suggestion.

We put all the Figures and Tables in the text to make the article read smoothly.
And will reconfirm the format of the article with the journal editor.

All the best.

Reviewer 3 Report

Dear Author,

thank you for modify your article according my suggestions. It seems to me that it has been improved.

Please pay attention to the format.

Best Regards

Author Response

Response to Reviewer 3:

Thanks for your suggestion.
We will reconfirm the format of the article with the journal editor.

All the best.

Round 3

Reviewer 2 Report

The figures and tables are a welcome addition to the paper.

The variables are well explained in the paper, although they are very different from usual variables (both old and young house for example).

With these changes the paper can be published.